# Clinical Course, Serologic Response, and Long-Term Outcome in Elderly Patients with Early Lyme Borreliosis

**DOI:** 10.3390/jcm7120506

**Published:** 2018-12-02

**Authors:** Katarina Boršič, Rok Blagus, Tjaša Cerar, Franc Strle, Daša Stupica

**Affiliations:** 1Department of Infectious Diseases, University Medical Center Ljubljana, Japljeva 2, 1525 Ljubljana, Slovenia; franc.strle@kclj.si; 2Institute for Biostatistics and Medical Informatics, Faculty of Medicine Ljubljana, Vrazov trg 2, 1104 Ljubljana, Slovenia; rok.blagus@mf.uni-lj.si; 3Institute of Microbiology and Immunology, Faculty of Medicine Ljubljana, Zaloška 4, 1000 Ljubljana, Slovenia; tjasa.cerar@mf.uni-lj.si; 4Faculty of Medicine Ljubljana, University of Ljubljana, Japljeva 2, 1525 Ljubljana, Slovenia

**Keywords:** erythema migrans, Lyme borreliosis, elderly, post-Lyme symptoms

## Abstract

Infected elderly people often present with signs and symptoms that differ from those in younger adults, but data on the association between patient age and presentation of early Lyme borreliosis (LB) are limited. In this study, the association between patient age (18–44 years, young vs. 45–64 years, middle-aged vs. ≥ 65 years, elderly) and disease course, microbiologic characteristics, and the long-term outcome of treatment was investigated prospectively in 1220 adult patients with early LB manifesting as erythema migrans (EM) at a single-center university hospital. Patients were assessed at enrolment and followed-up for 12 months. Age was associated with comorbidities, previous LB, presenting with multiple EM, and seropositivity to borreliae at enrolment. The time to resolution of EM after starting antibiotic treatment was longer in older patients. At 12 months, 59/989 (6.0%) patients showed incomplete response. The odds for incomplete response decreased with time from enrolment (odds ratio (OR) of 0.49, 0.50, and 0.48 for 2-month vs. 14-days, 6-month vs. 2-month, and 12-month vs. 6-month follow-up visits, respectively), but were higher with advancing age (OR 1.57 for middle-aged vs. young, and 1.95 for elderly vs. young), in women (OR 1.41, 95% confidence interval (CI) 1.01–1.96), in patients who reported LB-associated constitutional symptoms at enrolment (OR 7.69, 95% CI 5.39–10.97), and in those who presented with disseminated disease (OR 1.65, 95% CI 1.09–2.51). The long-term outcome of EM was excellent in patients of all age groups. However, older patients had slower resolution of EM and higher odds for an unfavorable outcome of treatment (OR 1.57, 95% CI 1.05–2.34 for middle-aged vs. young; and OR 1.95, 95% CI 1.14–3.32 for elderly vs. young), manifested predominantly as post-LB symptoms. The presence of LB-associated constitutional symptoms at enrolment was the strongest predictor of incomplete response.

## 1. Introduction

Lyme borreliosis (LB) is the most prevalent tick-borne disease in the northern hemisphere, and erythema migrans (EM) represents the most common clinical manifestation of this disease [1]. After being treated according to guidelines, the large majority of patients recover completely, but up to 10% report persistent subjective symptoms, termed post-LB symptoms [2]. Differences in the causative *Borrelia* species, burden of borreliae, and immune response, but not serologic response to infection, were found to influence the clinical course of LB [3,4]. Differences in immune response have been implicated in the pathogenesis of post-LB symptoms [3,4].

The global population is aging [5]. Many infections such as bacteremia or meningitis are both more frequent and more often associated with adverse outcomes in older individuals. Moreover, infected elderly persons often present with signs and symptoms that differ from those in younger adults [6]. Whether patient age is of relevance for the clinical course and outcome of LB, and through which mechanisms, has been poorly investigated [7].

We aimed to investigate how age is associated with the clinical, microbiologic, and long-term outcome of treatment parameters in adult patients with early LB.

## 2. Experimental Section

### 2.1. Setting and Participants

The 1220 patients included in this study were ≥18 years of age with EM defined according to European criteria [8], and were enrolled prospectively in five clinical trials between June 2006 and October 2015 at the Lyme borreliosis outpatient clinic, University Medical Center Ljubljana, Slovenia [2,4,9,10,11]. Patients were referred by their general practitioners, and only exceptionally by other specialists. EM was defined as an expanding red or bluish-red plaque, with or without central clearing, developing days to weeks after a tick bite or after exposure to ticks in an LB endemic region. For a reliable diagnosis, the erythema must reach ≥5 cm in diameter. If the diameter is smaller, a history of tick bites, a delay in appearance of at least two days, and an expanding erythema at the site of the bite are required [8]. Multiple EM is defined as the presence of two or more erythemas, at least one of which must fulfil the size criterion for solitary EM [12]. Patients with multiple EM were over-represented in our sample because they were enrolled preferentially in two of the clinical trials in the present analysis [10,11]; patients with previous LB were under-represented because such patients were excluded from two of the eligible trials [2,9]. Patients were categorized as young (18–44 years), middle-aged (45–64 years), or elderly (≥65 years) [13]. At enrolment, patients were prescribed antibiotic regimens in accordance with treatment guidelines [14], in accordance with individual study protocols. Thus, patients with solitary EM were treated with doxycycline for 10 days (107 patients, 8.8%), 14 days (400 patients, 32.8%), or 15 days (259 patients, 21.2%); or with cefuroxime axetil for 15 days (199 patients, 16.3%); or with amoxicillin for 15 days (60 patients, 4.9%). Patients with multiple EM received intravenous ceftriaxone (96 patients, 7.9%) or doxycycline (99 patients, 8.1%) for 14 days [2,4,9,10,11].

The study was approved by the Medical Ethics Committee of the Ministry of Health of the Republic of Slovenia (No. 0120/-670/2017/4) and registered at http://clinicaltrials.gov, identifier NCT03371563. All patients gave written consent for participation. 

### 2.2. Evaluation of Patients 

At baseline and at follow-up (14 days, then 2, 6, and 12 months), patients’ medical history was taken and physical examination performed. Additionally, patients were asked, without prompting, an open question about any health-related symptoms that had newly developed or worsened since the onset of EM. If these symptoms had no other known medical explanation, they were regarded as LB-associated constitutional symptoms at enrolment or post-LB symptoms at follow-up. On day 14, patients were asked about medication compliance. 

Complete response to treatment was defined as a return to pre-LB health status, partial response as the presence of post-LB symptoms, and treatment failure as the occurrence of new objective signs of LB, persistence of borreliae as detected by culture of a skin re-biopsy sample, persistence of EM at ≥2 months post-treatment, or any combination of the three. Persistence of EM was defined as EM that could still be seen in daylight and at room temperature. Failures and partial responses were categorized as incomplete response.

### 2.3. Microbiologic Methods

Serologic data were obtained using either indirect chemiluminescence immunoassay (IgM antibodies to OspC and VlsE, IgG antibodies to VlsE borrelial antigens; LIAISON, Diasorin, Italy), the C6 Lyme ELISA kit (IgM and IgG antibodies to C6 peptide derived from VlsE; Immunetics^®^, Oxford Immunotec, Marlborough, MA, USA), or an immunofluorescence assay with a local skin isolate of *B. afzelii* as an antigen [15]. Results were interpreted according to the manufacturers’ instructions or as titers, respectively; in the immunofluorescence assay, titers ≥ 1:128 were considered positive. 

At the baseline visit, a skin biopsy was taken at the expanding edge of the primary EM and a sample placed in 6 mL modified Kelly-Pettenkofer culture medium (MKP). If the first skin specimen was culture positive for borreliae, a second skin biopsy was collected from the same site 2–3 months after the first. Baseline blood samples were cultured for borreliae as previously described [16]. Cultures were examined weekly under dark-field microscopy for the presence of spirochetes, and were interpreted as negative if no growth was established by 9 weeks. Isolates were identified to the species level using pulsed-field gel electrophoresis after MluI restriction of genomic DNA [17], PCR-based restriction fragment length polymorphism of the intergenic region [18], or real-time PCR targeting the *hbb* (U48676.1) gene [19].

### 2.4. Statistical Methods

Categorical data were summarized as frequencies (%) and numerical data as medians (interquartile range, IQR). Differences between the age groups were tested using Kruskal–Wallis one-way analysis of variance or the chi-square test with Yates continuity correction. Post-hoc *p* values for all possible pairwise comparisons of the age groups were adjusted using the Holm’s method. The association between the proportion of patients with incomplete response and age at each follow-up time point was tested with the chi-square test with Yates continuity correction. 

The median duration of the EM was calculated using the Kaplan–Meier method; the log-rank test was used to test the difference between the duration curves in different age groups. The association between incomplete response and a predetermined set of covariates (patients’ sex, presence of comorbidities, presence of LB-associated constitutional symptoms at enrolment, presence of multiple EM, age, and time from enrolment) was estimated using multiple logistic regression. To account for multiple measurements in each patient and the fact that the patients were a part of five different trials, the analysis was also adjusted for a subject variable and trial variable as random effects. Results are presented as odds ratios (OR) with 95% confidence intervals (CI). R statistical language (version 3.4.1) was used for the analyses [20]. 

## 3. Results

Among the 1220 enrolled patients with EM, 369 were young (30.2%), 627 (51.4%) middle-aged, and 224 (18.4%) elderly (Table 1). Among the 224 elderly, 173 were 65–74 years old, 48 were 75–84 years, and 3 patients were ≥85 years old.

### 3.1. Pre-Treatment Characteristics

Patient sex and the presence of comorbidities were associated with age (Table 1). Post hoc *p* values for patient sex were <0.001, <0.001, and 0.952 for the comparison of young vs. middle-aged, young vs. elderly, and middle-aged vs. elderly, respectively, and <0.001, <0.001, and <0.001 for comorbidities. 

The associations between age and previous LB, as well as between age and the presence of multiple EM were significant (*p* < 0.002 and *p* < 0.001, Table 1). There were relatively more elderly patients in the group who had experienced previous LB than among patients without previous LB, and there were fewer elderly people among patients with multiple EM than in the group with solitary EM.

Overall, patients with multiple EM reported LB-associated constitutional symptoms at enrolment more often than those with solitary EM (92/195, 47.2% vs. 281/1025, 27.4%; *p* < 0.001). This was the opposite, although not significant, for elderly patients (2/18, 11.1% vs. 60/206, 29.1%; *p* = 0.167). At enrolment, older patients reported LB-associated constitutional symptoms as frequently as their younger counterparts (Table 1). 

### 3.2. Microbiologic Results According to Age 

At enrolment, elderly patients were more often seropositive for borreliae than were middle-aged or younger adults, but the difference was not significant when accounting for multiple comparisons (*p* = 0.027, Table 1) and was almost nonexistent at 12 months (*p* = 0.131). The seropositivity rate at enrolment was comparable between patients who were skin or blood culture positive (or both) and those who were culture negative: 230/377, 61.0% vs. 202/318, 63.5% using chemiluminescence immunoassay (*p* = 0.547); 10/91, 10.9% vs. 18/112, 16.1% using immunofluorescence assay (*p* = 0.401); and 47/63, 74.6% vs. 35/56, 62.5% when C6 Lyme ELISA was performed (*p* = 0.220).

Patients with history of LB were more often seropositive for borreliae than those who had not experienced LB in the past (62/87, 71.3% vs. 600/1105, 54.3%; *p* = 0.002), and patients with multiple EM were more often seropositive at enrolment than patients with solitary EM (133/189, 70.4% vs. 529/1003, 52.7%; *p* < 0.001).

*B. burgdorferi* sensu lato was isolated from pre-treatment skin biopsy specimens in 635/1142 (55.6%) patients who consented to the procedure, with comparable frequency in all three age groups. Among the isolates, 91.2% were *B. afzelii*, but the proportion of non-*afzelii* species increased with age. A re-biopsy was performed in 610/635 (96.1%) initially culture-positive patients and only one re-biopsy was culture positive. Blood cultures were positive only exceptionally (Table 1).

### 3.3. Treatment Outcome According to Age

All except 21 patients were re-examined at 14 days and all stated compliance with taking the study drug. The median time to resolution of EM after starting antibiotic treatment was 7 days in all age groups, but the distribution of time to resolution (IQR 4–10 days, 4.3–14 days, and 5–14 days in young, middle-aged, and elderly patients, respectively) showed significant prolongation with advancing age (*p* = 0.028).

The large majority (≥83%) of patients in all age groups showed complete response from 2 months onward, returning to their pre-LB health status. The proportion of patients with incomplete response, represented predominantly by the presence of post-LB symptoms, steadily decreased during follow-up in all age groups, differences between the groups reaching significance (according to univariate analysis) only at the 6-month visit (Table 2). However, the multivariate analysis, including age, time from enrolment, patient sex, presence of comorbidities, presence of multiple EM, and presence of LB-associated constitutional symptoms at enrolment, indicated that the odds for incomplete response decreased with time from enrolment, and were higher for older patients, women, patients with multiple EM, and even more so for those who presented with LB-associated constitutional symptoms at enrolment (Table 3).

At 12 months, 59/989 (6.0%) patients showed incomplete response. Findings were also similar for the last evaluable visit (85/1217, 6.9%). None of the patients with post-LB symptoms at 12 months qualified as having “post-Lyme disease syndrome”, because the reported symptoms did not necessitate reduction of their previous activity levels, as self-assessed by the patients [14].

Treatment failure was documented in 14 patients: two (0.2%) with solitary EM developed new objective manifestations of LB by the 14-day visit (one meningitis, the other multiple EM); one skin culture-positive patient with solitary EM had a positive skin culture for *B. afzelii* two months after completion of therapy, but these results were inconclusive and reinfection could not be ruled out; and in 11 patients (nine solitary EM, two multiple EM), residual erythema could still be seen at the 2-month visit. In all treatment failures, the patients were retreated with antibiotics recommended for early LB and had an uneventful further course.

## 4. Discussion

Our analysis of 1220 adult European patients with EM, 635 of whom were culture-positive, and who were stratified according to age into young, middle-aged, and elderly, showed several differences in clinical and microbiologic characteristics of the disease according to age, but excellent outcome in all age groups. Age was associated with comorbidities, previous LB, presenting with multiple EM, seropositivity to borreliae at enrolment, and with differences in the spectrum of the causative borreliae. 

Up until now, information on the impact of age on the clinical course and outcome of early LB was limited to one report from the U.S. comparing 283 culture-positive patients over the age of 50 with younger adults, 128 of whom had been followed-up for 11–20 years [7]. The authors found that older age at the time of diagnosis did not impact the initial clinical features or long-term outcome of this infection. 

The age distribution of our patients accords with epidemiologic characteristics of EM in Slovenia, which most frequently affects people aged 45–64 years [21]. Overall, female predominance among our patients with EM (675/1220, 55.3%) is consistent with reports from other European countries [22]; however, the sex distribution of patients according to their age did not reflect the sex ratio in the general population, which might be related to differences in social behavior between women and men of different ages—the proportion of females was lower among the young patients and higher in middle-aged patients than in the general Slovenian population of the same age (42.0% vs. 48%, *p* = 0.024; and 61.2% vs. 49.4%, *p* < 0.001, respectively), but comparable for the elderly (60.7% vs. 60.3%, *p* = 1.0).

Lyme borreliosis is endemic in Slovenia [20], therefore it was not surprising to find that 91/1220 (7.5%) patients with EM reported previous LB. Moreover, this proportion would be higher (175/1304, 13.4%) if patients with a history of LB had not been excluded from two [2,9] of the five studies that formed the basis of our analysis. Multiple as opposed to solitary EM occurs more commonly in children than in adults [23,24,25], and multiple EM was found more often in younger patients in our previous report comparing adults with multiple and solitary EM [11]. The association between age and multiple EM was further substantiated in the present analysis. We do not have a conclusive explanation for this finding, however the higher proportion of elderly patients who tested seropositive at enrolment might be implicated, assuming that previous infection with borreliae offers some protection against disseminated disease. An association between age and borrelial seroprevalence has been established in healthy blood donors [26], presumably because the likelihood of encountering borrelial infection in an endemic environment increases with age. 

Comparison of patients with multiple vs. solitary EM in our previous study [11], as well as in the present analysis has shown that patients with disseminated disease present with LB-associated constitutional symptoms more often. However, in the present study we found that older patients reported LB-associated constitutional symptoms at enrolment as frequently as their younger counterparts, even though the proportion of elderly patients was lower among the patients with multiple EM as opposed to solitary EM. It could be that the effect of specific protective immunity counterbalanced the effect of age-associated pathophysiologic alterations associated with the higher frequency of comorbidities, some of which might elicit or aggravate LB-associated symptoms. 

The time to resolution of EM after starting antibiotic treatment showed significant prolongation with advancing age (*p* = 0.028), which accords with delayed response to therapy in elderly patients with other types of infection, such as pneumonia [6,27]. 

In the univariate analysis, older age was associated with an unfavorable outcome of treatment only at the 6-month visit. However, the multivariate analysis showed that the odds for incomplete response, manifested predominantly by persistence of post-LB symptoms, were associated with higher age, female sex, presence of disseminated disease, and presence of LB-associated constitutional symptoms at enrolment, the latter showing the strongest effect. Yet in none of the patients were there post-LB symptoms of sufficient severity to be functionally disabling. These excellent outcome results, including the potentially more fragile subpopulation of elderly patients, suggest that the existence of the entity “post-Lyme disease syndrome”, as defined by the Infectious Diseases Society of America [14], is extremely rare, at least for adequately treated patients with EM in Europe. The term “post-LB symptoms” appears to encompass an unfavorable treatment outcome in a small minority of patients with EM.

Treatment failure was documented in only 14 (1.1%) patients. In the 11 cases with persisting EM, it is questionable whether these cases truly represented treatment failures, because pale residua at the location of previous EM may have represented post-inflammatory skin hyperpigmentation and there was no supporting evidence for the presence of an active infection in these patients at follow-up. The twelfth case was regarded as treatment failure because *B. afzelii* was cultured from a re-biopsy skin specimen. The strain of *B. afzelii* that grew however, differed from the original *B. afzelii* isolate when the plasmid profiles of the two strains were compared. Although we interpreted this case as treatment failure, on the assumption that the original infection was mixed [28], there is also the possibility that the post-treatment culture became contaminated in the laboratory or that the patient experienced a new (asymptomatic) infection. Disease progressed during antibiotic therapy in only two patients with solitary EM—one developed meningitis, the other multiple EM. They were both retreated and experienced uneventful outcomes.

In Slovenia and in wider Europe, *B. afzelii* is by far the most prevalent causative species of LB skin manifestations. This has been shown for early skin involvement (EM), but also for chronic atrophic acrodermatitis, even though the latter is more likely to be diagnosed in older individuals [29]. We do not have a reliable explanation for the finding that other *Borrelia* species in addition to *B. afzelii* were more frequent in older patients in the present analysis, and can only speculate that specific pathogen-host interaction mechanisms, which might be age-related, were involved. Indeed, Nadelman et al. showed that reinfections with *B. burgdorferi* were due to strains that differed from those causing the primary infection in 22 paired episodes of EM, leading the authors to conclude that this shift in the profile of the causative agents could be associated with partial protective immunity gained from the previous infection [30]. The same mechanism may account for the smaller proportion of older patients infected with *B. afzelii* in the present study.

Our study has several limitations. First, some of the post-LB symptoms in the patients could have been wrongly attributed to LB because of their non-specific nature, but some could have been missed. Second, the post-LB symptoms and their potential impact have not been assessed by any validated measure of functioning or quality of life, yet objective measures would be preferable in the future. Third, we could not exclude the possibility that the use of an open-ended question technique had an impact on symptoms reporting, which may have been influenced also by age or sex. Fourth, patients with a history of LB were under-represented and patients with multiple EM over-represented in the analysis, however we recognized this potential bias and adjusted the analysis accordingly. Fifth, our results are applicable to other European regions with similar ratios of *Borrelia* genospecies causing EM, but may not entirely apply to North America, where LB is caused nearly exclusively by *B. burgdorferi* sensu stricto [14]. 

Due to analyzing a high number of patients with systematically collected clinical and microbiological data, which were the strengths of our study, we were able to identify some important age-related characteristics of LB, which give further insight into the pathogenesis of this disease. 

In conclusion, in our study of European patients with EM, most of whom were infected with *B. afzelii*, the rate of disseminated disease and the prevalence of *B. afzelii* as the causative species were lower in older patients, which might be related to the higher rate of previous infection and consequential seroreactivity, representing partial immunity in older age. Erythema resolved more slowly in the older patients. The treatment outcome was excellent in all age groups, nevertheless higher age, female sex, the presence of disseminated disease, and LB-associated constitutional symptoms at enrolment were associated with incomplete response manifested predominantly as post-LB symptoms.

## Figures and Tables

**Table 1 jcm-07-00506-t001:** Pre-treatment characteristics of patients with erythema migrans according to age.

Characteristic	Young	Middle-Aged	Elderly	*p* Value ^a^
	369 (30.2%)	627 (51.4%)	224 (18.4%)	
Male sex	214 (58)	243 (38.8)	88 (39.3)	<0.001
History of Lyme borreliosis				0.002
Yes	17 (14.2)	72 (60)	31 (25.8)	
No	352 (32)	555 (50.5)	193 (17.5)	
Comorbidities ^b^	54 (14.6)	298 (47.5)	171 (76.3)	<0.001
Tick bite ^c^	179 (48.5)	319 (50.9)	117 (52.2)	0.642
Days since EM first observed	10 (5–24)	11 (5–25)	10 (4–28)	0.835
Diameter of primary EM, cm	15 (9–22)	14 (9–20)	14 (10.8–24)	0.194
EM with central clearing ^d^	205 (55.6)	288 (45.9)	112 (50.0)	0.013
Multiple EM				<0.001
Yes	80 (41.6)	97 (49.7)	18 (9.0)	
No	289 (28.2)	530 (51.7)	206 (20.1)	
LB-associated constitutional	121 (32.8)	190 (30.3)	62 (27.7)	0.415
Symptoms ^e^				
Fatigue	53 (14.4)	98 (15.6)	33 (14.7)	0.853
Arthralgia	45 (12.2)	67 (10.7)	20 (8.9)	0.457
Headache	47 (12.7)	81 (12.9)	26 (11.6)	0.877
Myalgia	18 (4.9)	55 (8.8)	14 (6.3)	0.060
Seropositive ^f^	191/358 (53.4)	331/614 (53.9)	140/220 (63.6)	0.027
Skin culture positive	187/340 (55.0)	331/588 (56.3)	117/214 (54.7)	0.888
*B. afzelii*	177 (94.7)	296 (89.4)	95 (81.2)	0.004 ^g^
*B. garinii*	7 (3.7)	19 (5.7)	14 (12.0)	
*B. burgdorferi* sensu stricto	1 (0.5)	5 (1.5)	5 (4.3)	
Unidentified ^h^	2 (1.1)	11 (3.3)	3 (2.6)	
Blood culture positive	5/340 (1.5)	7/614 (1.1)	6/220 (2.7)	0.244

Data are median (interquartile range, IQR), n (%), or n/n (%). Abbreviations: EM, erythema migrans; LB, Lyme borreliosis. ^a^ Overall *p* value for comparisons between all age groups. Because of multiple comparisons, *p* < 0.01 was considered significant. ^b^ Patients with underlying chronic illness such as arterial hypertension, hyperlipidemia, osteoporosis, diabetes mellitus, thyroid disease, cardiac rhythm abnormality, psychiatric illness, ischemic heart disease, osteoarthritis, or asthma. ^c^ Patients with a history of tick bites at EM site. ^d^ In patients with multiple EM, the primary lesion was assessed for central clearing. ^e^ Patients who reported constitutional symptoms that had newly developed or worsened since the onset of the EM. Some patients had more than one constitutional symptom. ^f^ Positive test result for IgM and/or IgG antibodies to *B. burgdorferi* sensu lato at enrolment using chemiluminescence immunoassay (551/852, 64.7%), immunofluorescence assay (29/221, 13.1%), and the C6 Lyme ELISA (82/119, 68.9%). ^g^ Comparison between *B. afzelii* and other identified *Borrelia* species. ^h^ Borrelia species identification was not possible due to slow-growing isolates or visible but not growing strains.

**Table 2 jcm-07-00506-t002:** Number (%) of patients with erythema migrans who had incomplete response at follow-up visits according to age.

	All	Young	Middle-Aged	Elderly	*p* Value ^a^
	*n* = 1220	*n* = 369	*n* = 627	*n* = 224	
14 days post-enrolment	258/1199 (21.5)	76/362 (21.0)	135/616 (21.9)	47/221 (21.3)	0.940
2 months post-enrolment	172/1176 (14.6)	41/351 (11.7)	95/605 (15.7)	36/220 (16.4)	0.171
6 months post-enrolment	101/1032 (9.8)	17/291 (5.8)	61/544 (11.2)	23/197 (11.7)	0.028
12 months post-enrolment	59/977 (6.0)	10/271 (3.7)	34/513 (6.6)	15/193 (7.8)	0.137
Last evaluable visit	85/1217 (6.9)	19/368 (5.2)	49/625 (7.8)	17/224 (7.6)	0.258

^a^ Overall *p* value for comparisons between age groups was estimated using the normal approximation with continuity correction. *p* < 0.05 was considered significant.

**Table 3 jcm-07-00506-t003:** Factors associated with incomplete response according to age (middle-aged versus young and elderly versus young).

	OR (95% CI) ^a^	*p* Value ^b^
Age		0.038
Middle-aged vs. young	1.57 (1.04–2.37)	0.031
Elderly vs. young	1.94 (1.12–3.37)	0.018
Time from enrolment		<0.001
2 months vs. 14 days	0.48 (0.37–0.63)	<0.001
6 vs. 2 months	0.50 (0.36–0.68)	<0.001
12 vs. 6 months	0.47 (0.32–0.70)	<0.001
Sex (female vs. male)	1.43 (1.01–2.02)	0.041
Presence of comorbidities	0.85 (0.59–1.24)	0.399
Multiple EM vs. solitary EM	1.67 (1.08–2.58)	0.022
Presence of LB-associated constitutional symptoms at enrolment (yes vs. no)	8.47 (5.79–12.38)	<0.001

Abbreviations: OR, odds ratio for incomplete response; CI, confidence interval; EM, erythema migrans; LB, Lyme borreliosis. ^a^ Estimated from a multiple logistic regression model with incomplete response as the dependent variable, adjusted for patient effect. Each OR is adjusted for all other variables in the table. ^b^
*p* < 0.05 was considered significant.

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
