# Peer review of "Clinical Course, Serologic Response, and Long-Term Outcome in Elderly Patients with Early Lyme Borreliosis"

_jcm, 2018, doi:10.3390/jcm7120506_

Reviewer 1 Report

Borsic et al. have described a post-hoc analysis on the course of early Lyme borreliosis in patients enrolled into 5 prospective clinical trials that were previously published. For this particular analysis, they have focused on the differences between age groups.

The paper is very well written, and the clinical assessments of patients are excellent. 

I do have several remarks:

1. A relatively large proportion of patients (20%) had multiple EM (MEM). While the authors acknowledge that this has been due to inclusion of specific populations from MEM trials, this may lead to known and unknown confounders. E.g., as patients with MEM tend to be younger, the case mix in terms of age distribution has changed, which may be a significant limitation in a study specifically describing age differences. Likewise, the distribution of non-afzelii strains varied by age group, which may also have been impacted by the MEM overrepresentation in the younger age groups. These, and other unknown confounders may not have fully accounted for in the regression analysis. A sensitivity analysis excluding the MEM patients may attempt to explore these issues.

2. The division into 3 age groups, with 45y and 65y as limits, seems arbitrary, and the middle-age group is almost 3 times larger than the older-age group. Why were these age limits chosen? The authors may want to think about increasing the power of the study when age groups are equally sized.

3. For the multivariate analysis, sensitivity likely is lost when applying 3 separate categorical age groups. When investigating the effect of age on presentation and outcomes, applying age as a continuous variable would make more sense, and increase the power to detect any clinically important determinants. This seems to be a crucial methodological issue.

4. Interestingly, the authors state that at 12 months, 6% of patients showed incomplete response, but none of the patients qualified as having PTLDS, as their activity levels were not affected. This would require further elucidation. Was impairment of activity systematically measured, and if so, on which scale? Where there any objective measures of activity and impairments of daily life? This is important, as others have claimed a substantial proportion of patients with significant impairments after 12 months. The authors may want to speculate in the discussion why their patients did not develop PTLDS. Are there cultural differences between countries, or may there have been an incomplete assessment of QoL?

5. Minor suggestion: Please clarify “intercept” in Table 3.

Author Response

Author's reply to the Review Report (Reviewer 1)

Comments and Suggestions for Authors

Borsic et al. have described a post-hoc analysis on the course of early Lyme borreliosis in patients enrolled into 5 prospective clinical trials that were previously published. For this particular analysis, they have focused on the differences between age groups.

The paper is very well written, and the clinical assessments of patients are excellent. 

I do have several remarks:

1. Reviewer: A relatively large proportion of patients (20%) had multiple EM (MEM). While the authors acknowledge that this has been due to inclusion of specific populations from MEM trials, this may lead to known and unknown confounders. E.g., as patients with MEM tend to be younger, the case mix in terms of age distribution has changed, which may be a significant limitation in a study specifically describing age differences. Likewise, the distribution of non-afzelii strains varied by age group, which may also have been impacted by the MEM overrepresentation in the younger age groups. These, and other unknown confounders may not have fully accounted for in the regression analysis. A sensitivity analysis excluding the MEM patients may attempt to explore these issues.

Answer: We agree with the reviewer that the unequal sampling of patients with MEM could have biased our results if we had not adjusted the analysis appropriately, which we believe we had by including dissemination (MEM vs solitary EM) as a covariate in the multivariate model. As suggested by the reviewer, we added the trial variable as another random effect to account for some other potential biases which could arise from the fact that the data come from different trials. Please, see lines 120-125.

2. Reviewer:  The division into 3 age groups, with 45y and 65y as limits, seems arbitrary, and the middle-age group is almost 3 times larger than the older-age group. Why were these age limits chosen? The authors may want to think about increasing the power of the study when age groups are equally sized.

Answer: The age distribution in our study reflects the age distribution of EM in the general population since Lyme borreliosis has been a mandatorily reportable disease in Slovenia since 1986 and the data on the incidence of EM in Slovenia is publicly available. Since there is no universe definition of old because it differs according to the context, we decided to  determine age categories based on conclusions drawn from the available age categorizations (WHO, Lee CC, Wang JL, Lee CH, et al. Age-Related Trends in Adults with Community-Onset Bacteremia. Antimicrob Agents Chemother. 2017 Nov 22;61(12)). Information was added to the manuscript. Please, see line 69.

We agree with the reviewer, that having three groups of same size would increase the power of the study, but since the determination of the categories would in this case be data driven, this could potentially mean that we would obtain results leading to too optimistic conclusions, therefore we decided not to apply such an approach.

3. Reviewer: For the multivariate analysis, sensitivity likely is lost when applying 3 separate categorical age groups. When investigating the effect of age on presentation and outcomes, applying age as a continuous variable would make more sense, and increase the power to detect any clinically important determinants. This seems to be a crucial methodological issue.

Answer: We thank the reviewer for this comment because we had the same dilemma when planning the analysis. However, we decided not to apply age as a continuous variable because by applying such an approach the obtained results would be in a form that would be much more difficult to understand for a clinical doctor with average statistical knowledge, which was the reason to abandon this alternative.  Treating age as continuous predictor instead of a categorical one in the multivariate analysis would mean that splines or similar would have to be used due to a non-linear association. This would then also mean that age would have to be considered as a continuous variable throughout the analysis which would complicate the presentation substantially.

4. Reviewer:  Interestingly, the authors state that at 12 months, 6% of patients showed incomplete response, but none of the patients qualified as having PTLDS, as their activity levels were not affected. This would require further elucidation. Was impairment of activity systematically measured, and if so, on which scale? Where there any objective measures of activity and impairments of daily life? This is important, as others have claimed a substantial proportion of patients with significant impairments after 12 months. The authors may want to speculate in the discussion why their patients did not develop PTLDS. Are there cultural differences between countries, or may there have been an incomplete assessment of QoL?

Answer: We referred to the definition of post-Lyme disease syndrome as it is defined by the Infectious Diseases Society of America (Ref No 13). In regard to severity of symptoms, this definition defines severity of symptoms as follows: “subjective symptoms are of such severity that, when present, they result in substantial reduction in previous levels of occupational, educational, social, or personal activities”, which was not the case in any of our patients. The activity levels were not specifically measured, still we believe our conclusion is reasonable because none of the patients made any change in his/her occupational, educational, social, or personal activities due to post-LD symptoms. We amended the manuscript as suggested by the reviewer. Corrections done. Please, see line 271, 373-377.

5. Reviewer: Minor suggestion: Please clarify “intercept” in Table 3.

Answer: As suggested by one of the reviewers the results for the intercept have been removed. Please, see line 250.

Reviewer 2 Report

The study describes serologic response, duration and outcome of an erythema migrans in elderly patients (>64 years) compared to two groups of younger patients. The study has a large sample size of over 1200 patients with an EM. The study is new and of interest to the field of Lyme borreliosis, as there is very little known on LB in elderly.

Major comments:

How do you think the fact that patients were included from 5 trials with differenct inclusion criteria, treatment and diagnostic methods influenced the results on course and outcome of an EM? Possible also the different diagnostic test used influence seropositivity? Is there a way to some how correct for this in data analysis?

The results section is difficult to follow. The paper would benefit if you compare elderly (>64 years) with younger patients (<65 years). Please also add the age distribution of the elderly. The aim of the study is to investigate how age influences LB. Why is not in the results of logistic regression age the main outcome measure instead of incomplete response? Now in univariate analyses age is the dependent outcome measure. No multivariate analyses on this is presented. And then multivariate analyses are presented with as outcome measure incomplete response, and no univariate results are presented on this.

Abstract

The objective is to look at age and Lyme borreliosis. Why is age not the dependent outcome measure in logistic regression analysis?

Line 24: The time to… Can you give the median time in each of the age groups to support such a conclusion.

There is nothing mentioned in the abstract on data analysis and what kind of statistics is used and which groups are exactly compared.

It is not necessary to report P-values and OR with 95% confidence interval, p-values can be deleted.

Also give the OR for age and time from enrollment. You can categorize age in 10 years, and possibly also make categories for time from enrollment.

Introduction

Line 39: an erythema migrans

Line 40: recommended as according to guidelines??

Line 40: the large majority. Can you give %? How many percent report persistent symptoms? How were post-LB symptoms defined?

Line 48: is implicated.. is of relevance for the clinical course

Line 50: delete word “Here”

Line 51: delete word “parameters”

 Methods

That the five clinical trials all have their own inclusion criteria, diagnostic methods, treatment can influence results. This is a limitation that should be mentioned in the discussion. Please also give some implications for your results.

Line 62: Patients with multiple EM were over-represented and patients with previous LB underrepresented. This can be put in the discussion section under limitations. What are the implications? Now it is only mentioned in discussion without possible implications given. Possibly there is also some way to correct for this in data-analysis?

Line 66: Elderly are defined as ≥65 years. The changes in immune response is by 75 and over, and 65 and over are not considered elderly in geriatrics. In results add age distribution for elderly: how many 65+, 75+, 75+

Line 71: Different treatments were non-inferior to comparators. What exactly is meant with this? That the different treatment regimes did not influence study results? The study looks at outcome and duration of EM. It is likely that this is influenced by treatment regime. Why do you think this is not the case? Is there a possibility to correct in multivariate analyses for different treatments?

Line 83: how were post-LB symptoms defined and assessed? As it is now described, patients had to have post-LB symptoms in order to have incomplete response? If a patients has new objective signs for LB without post-LB symptoms, what is done?

Line 94: Is the titer above 1:128 considered positive, is this what is recommended by manufacturer?

Line 106: Is the Yates continuity correction really needed? Is that not only when the expected cell count is below 10. With the large sample population you would expect this is not the case.

Overall, a lot of conservative statistical methods. Are all these corrections needed? The Holm method to adjust for multiple testing. Were there p-values that were first significant and after correction not anymore? Also multiple logistic regression was done to correct for clustering of data within patients. Again, do results change if you do not correct and is it really necessary?

Line 108: what is meant with the normal approximation?

The median duration of EM was calculated using survival methods. If you know the duration of the EM in days of each patient. Is it not more correct to compare median number of days in each group? What is the benefit of using survival analysis? Also as in the results no survival curves are shown and only one P-value is given. Better to use Mann-Whitney U-test to compare two median number of days?

 Results

Line 131: what were the comorbidities and how were these defined and assessed?

Line 154. “Because of multiple comparisons, P<0.1 was considered significant.” This is not in methods. In methods you say you correct for multiple testing with Holm correction.

Please avoid comparison of study results with other studies in the results section. This should only be done in the discussion. This concerns Line 120-121.

Line 166: the fact that you compare to the general Slovenian population is not mentioned in methods. Please put this in discussion.

Line 169-170: this is already mentioned in methods. Should be in discussion. Not in results and methods.

Line 178: please delete the word “trend”

Line 184: is this the corrected P-value?

Line 190: “comparably often in all age groups” This is not correct English

Line 199: Kaplan Meier P-value? No survival curves are shown. Please see my suggestion for statistical analysis.

Line 200: the large majority: what percent?

Line 201: What is the proportion of patients with incomplete response?

Line 204: It is incorrect to say the model was adjusted for.. the model included all these variables. However, in the methods it says that the model was adjusted for age and time for enrollment. How was the model constructed? Was forward or backward selection performed? Or did you just enter all variables of interest? Please consider taking eldery vs other ages as main outcome measure. Table 1 shows univariate analyses of factors associated with age, why did you not perform multivariate analyses on this? Why choose for factors associated with incomplete response only multivariate and not univariate?

Line 227: you do not need to put intercept in the table.

Line 231: Time from enrollment

Line 239: part of table 3 is missing.

Line 245-247: …but none qualified as having “post-Lyme disease syndrome”’, because the reported symptoms did not necessitate reduction of their previous activity levels. This definition should be in the methods. And how was it assessed whether there was a reduction in activity levels.

Line 252: Possibly in some patients it was not treatment failure but a reinfection. This is noted in the discussion, but could also be mentioned here that there was one case with reinfection, as this is a result.

How many patients had chemiluminescence immunoassay or the C6 ELISA, or immunofluorescence assay? Were ELISA results confirmed with immunoblot according to guidelines?

 Discussion

Please start the discussion with a summary of your main conclusion. Line 256 is a reference to another paper. Comparison with study results is for second paragraph. Line 260: What are the main differences with respect to age that you found?

Line 278: Possibly in elderly also the time between onset of EM and treatment is longer, which might partly explain the longer duration of resolution of EM. In patients aged 85 and over you would already expect patients not to be the same as before the event, taking into account comorbidities.

Line 281:

Strenghts of the study are missing in the discussion

Line 318 and 325: rephrase analysis into study

Line 333: this part focuses on post-LB symptoms. The aim of the study was to look at age.

Author Response

Author's reply to the Review Report (Reviewer 2)

Comments and Suggestions for Authors

The study describes serologic response, duration and outcome of an erythema migrans in elderly patients (>64 years) compared to two groups of younger patients. The study has a large sample size of over 1200 patients with an EM. The study is new and of interest to the field of Lyme borreliosis, as there is very little known on LB in elderly.

 Major comments:

1. Reviewer: How do you think the fact that patients were included from 5 trials with differenct inclusion criteria, treatment and diagnostic methods influenced the results on course and outcome of an EM? Possible also the different diagnostic test used influence seropositivity? Is there a way to some how correct for this in data analysis?

Answer: As suggested by the reviewer, we made corrections in the analysis approach. We additionally accounted for the fact that patients from 5 studies were included in the multivariate model by adding a random effect for the trial which in the multivariate analysis adjusts for these differences. The Methods section has been changed accordingly. Please, see lines 122-125, and Table 3, lines 251-261. Note that since all multiple EM patients arise from one trial, including trial as a fixed effect was not possible.

Furthermore:

- the main inclusion criteria for the 5 studies were practically the same and differed in two parameters, which we were aware of: one study preferentially included patients with multiple erythema migrans, and in two studies patients with a history of Lyme borreliosis in the past were excluded. We recognized that the uneven sampling might potentially bias our results and adjusted the analysis accordingly, nevertheless we recognized this as one of the study limitations. Please see lines 374-377.

- we did not include treatment regimen as a variable with potential influence on the course and outcome because all patients were treated with antibiotic regimens, which are recommended as efficacious treatment options for treating erythema migrans by the Infectious Diseases Society of America (Ref No 13). Nevertheless, we wrote that different treatment regimens used in the 5 studies were non-inferior to comparators. Corrections done. Please, see lines 75-76.

- the differences in diagnostic methods could not have influenced the results on the course and outcome of EM because no such differences existed. Truly, different serological methods were applied during the study period, however serology was not used as diagnostic criteria. Erythema migrans was defined according to European criteria which do not rely on serological test result. Please, see lines 58-60 and lines 60-61.

- different serological tests did influence serological results, and we added this information to the manuscript. However, since seropositivity was not used as diagnostic criteria in our study we believe these potential differences did not impact our results on the course and outcome. Furthermore, because each serological method was applied regardless of patients’ age, we believe that differences in serological methods did not impact the direction and strength of correlation between patients’ age and serological results, which was the main question regarding seropositivity in our study. Corrections done. Please, see lines 173-175.

 2. Reviewer: The results section is difficult to follow. The paper would benefit if you compare elderly (>64 years) with younger patients (<65 years). Please also add the age distribution of the elderly. The aim of the study is to investigate how age influences LB. Why is not in the results of logistic regression age the main outcome measure instead of incomplete response? Now in univariate analyses age is the dependent outcome measure. No multivariate analyses on this is presented. And then multivariate analyses are presented with as outcome measure incomplete response, and no univariate results are presented on this.

Answer: From statistical point of view, we believe we chose the right approach (where age always is a categorized independent variable) for the following reasons:

-in our study age was not the outcome of the study but a possible predictor of our primary outcome which was the incomplete response to treatment, hence it would be wrong to model age as the outcome for this purpose. Also, in other results (Table 1), age is always the predictor variable.

-it is not the goal of the study to “investigate how age influences LB” but how some aspects of LB are associated with age. Since this is an observational study, studying causal inference, as implied by “influences” is currently unfeasible unless when relying on some (strong) assumptions which cannot be verified from the data (e.g. the latent variable approach).

- our main question could also have been answered using the statistical approach (not the one proposed by the reviewer) which models age as a continuous (independent) variable (due to a highly non-linear relation modelled via splines or similar), however we did not decide to use such an approach because by applying it the obtained results would have been in a form that would have been much more difficult to understand for a clinical doctor with average statistical knowledge, which was the reason to abandon this alternative. We acknowledge that our approach can suffer from some potential loss of information, but considering only two groups (which we agree would significantly simplify the presentation) would in our opinion lead to too much loss of information.

 Abstract

1. Reviewer: The objective is to look at age and Lyme borreliosis. Why is age not the dependent outcome measure in logistic regression analysis?

Answer: Please see answer to Reviewer’s Major comment, No 2.  

2. Reviewer: Line 24: The time to… Can you give the median time in each of the age groups to support such a conclusion.

Answer: Since the median time to resolution of EM after starting antibiotic treatment was 7 days in all age groups, we think it is better not to add it into Abstract because the explanation for the significant difference is to long:” the distribution of time to resolution (IQR 4–10 days, 4.3–14 days, and 5–14 days in young, middle-aged, and elderly patients, respectively) showed significant prolongation with advancing age (P=.028)” as is explained in the Results, lines 216-220.

3. Reviewer: There is nothing mentioned in the abstract on data analysis and what kind of statistics is used and which groups are exactly compared.

Answer: Statistical methods applied in our study were rather complex and they were left out from the Abstract due to limitation of space and because this was not obligatory. However, the age groups compared in our study were already defined in the Abstract. Please, see lines 18-19.

4. Reviewer: It is not necessary to report P-values and OR with 95% confidence interval, p-values can be deleted.

Answer: Corrections done. Please, see lines 26, 28-31, and 33-34.

5. Reviewer: Also give the OR for age and time from enrollment. You can categorize age in 10 years, and possibly also make categories for time from enrollment.

Answer: Corrections done. Please, see lines 26-28.

 Introduction

1. Reviewer: Line 41: an erythema migrans

Answer: We believe our choice was correct because the manuscript has been lectured by a native English speaker, therefore we would rather not make corrections as suggested.

2. Reviewer: Line 40: recommended as according to guidelines??

Answer: Corrections done. Please, see line 42.

3. Reviewer: Line 40: the large majority. Can you give %? How many percent report persistent symptoms? How were post-LB symptoms defined?

Answer: Corrections done. Please, see line 43. We believe adding definition of post-LB symptoms at this point would lead to unnecessary repeating, because the definition is provided in the Methods section.

4. Reviewer: Line 48: is implicated.. is of relevance for the clinical course

Answer: Corrections done. Please, see line 50.

5. Reviewer: Line 50: delete word “Here”

Answer: Corrections done. Please, see line 52.

6. Reviewer: Line 51: delete word “parameters”

Answer: We believe “parameters” should not be deleted because “parameters” in this context refer also to adjectives “clinical, microbiological”, and not only to “long-term outcome of treatment”.

 Methods

1. Reviewer: That the five clinical trials all have their own inclusion criteria, diagnostic methods, treatment can influence results. This is a limitation that should be mentioned in the discussion. Please also give some implications for your results.

Answer: Please, see answer to Reviewer’s Major comment, No 1.

2. Reviewer: Line 62: Patients with multiple EM were over-represented and patients with previous LB underrepresented. This can be put in the discussion section under limitations. What are the implications? Now it is only mentioned in discussion without possible implications given. Possibly there is also some way to correct for this in data-analysis?

Answer: We agree with the reviewer that the unequal sampling of patients with MEM could have biased our results if we had not adjusted the analysis appropriately, which we believe we had by including dissemination (MEM vs solitary EM) as a covariate in the multivariate model. As suggested by the reviewer, we added the trial variable as another random effect to account for some other potential biases which could arise from the fact that the data come from different trials. Please, see lines 122-125 and lines 185-188. We also listed this as a limitation to our study. Please, see lines 377-380.

3. Reviewer: Line 66: Elderly are defined as ≥65 years. The changes in immune response is by 75 and over, and 65 and over are not considered elderly in geriatrics. In results add age distribution for elderly: how many 65+, 75+, 75+

Answer: We agree with the reviewer that there is not a universal definition of old because it differs according to the context. We amended the manuscript as proposed by the reviewer. Please, see lines 130-131.

4. Reviewer: Line 71: Different treatments were non-inferior to comparators. What exactly is meant with this? That the different treatment regimes did not influence study results? The study looks at outcome and duration of EM. It is likely that this is influenced by treatment regime. Why do you think this is not the case? Is there a possibility to correct in multivariate analyses for different treatments?

Answer: Please see answer to Reviewer’s Major comment, No 1. We decided to delete this sentence because it may cause unneeded dilemma. Corrections done. Please, see line 75-76.

5. Reviewer: Line 83: how were post-LB symptoms defined and assessed? As it is now described, patients had to have post-LB symptoms in order to have incomplete response? If a patients has new objective signs for LB without post-LB symptoms, what is done?

Answer: Corrections done. The definition of post-LB symptoms is provided in Methods. Please, see lines 84-86. Since failures and partial responses were categorized as incomplete response (lines 92-93), a patient with new objective signs of LB (failure) was categorized as incomplete response, regardless of the presence or absence of post-LB symptoms (partial response).

6. Reviewer: Line 94: Is the titer above 1:128 considered positive, is this what is recommended by manufacturer?

Answer: Immunofluorescence assay is an in-house test with local strain B. afzelii used as an antigen. As Lyme borreliosis is endemic in Slovenia, cut-off value of the test is based on findings in the control group from the same geographic region.

7. Reviewer: Line 106: Is the Yates continuity correction really needed? Is that not only when the expected cell count is below 10. With the large sample population you would expect this is not the case.

Answer: Indeed, Yates correction is asymptotically negligible, but it is not wrong to use it in this case. The role of the correction is to approximate a discrete distribution as given by the data by a continuous one as assumed by the chi-square distribution. The approximation, even with a large sample size, improves when using the Yates correction.

8. Reviewer: Overall, a lot of conservative statistical methods. Are all these corrections needed? The Holm method to adjust for multiple testing. Were there p-values that were first significant and after correction not anymore? Also multiple logistic regression was done to correct for clustering of data within patients. Again, do results change if you do not correct and is it really necessary?

Answer: As it was stated in the Methods section, Holm was used for post-hoc comparisons, i.e. all possible pairwise comparisons of the three age groups. This is (one of) the possible non-parametric equivalents to using Tukey’s test for post-hoc comparisons.  In the logistic regression model (in fact in any generalized linear model with a non-linear link function) not accounting for the clustering can not only bias the standard errors but it can also bias the point estimates. Hence, we think the methods that were used are correct.

9. Reviewer: Line 108: what is meant with the normal approximation?

Answer: We would like to thank the reviewer for spotting this. This was an error. The association between the incomplete response and age was also tested by using the chi-square test with the continuity correction. The Methods section has been changed accordingly. Please, see lines 114-117.

10. Reviewer: The median duration of EM was calculated using survival methods. If you know the duration of the EM in days of each patient. Is it not more correct to compare median number of days in each group? What is the benefit of using survival analysis? Also as in the results no survival curves are shown and only one P-value is given. Better to use Mann-Whitney U-test to compare two median number of days?

Answer: For some patients (11 in total) we did not have the exact information on the duration of EM, and we only knew that the duration was longer than 60 days, hence we believe using survival methods is the most appropriate way in this case.

 Results

1. Reviewer: Line 131: what were the comorbidities and how were these defined and assessed?

Answer: The information about comorbidities was obtained at enrolment when patients were asked about their past medical history which also contains a question about regular medications.

Corrections done. Please, see lines 81-82. The comorbidities are listed in the footnote of Table 1. Please, see lines 166-169.

2. Reviewer: Line 154. “Because of multiple comparisons, P<0.1 was considered significant.” This is not in methods. In methods you say you correct for multiple testing with Holm correction.

Answer: Holm was used to adjust the post-hoc p-values. What we report in the tables are not post-hoc but overall p-values. To account for multiple comparisons we decided to use a simple Bonferroni-type correction by adjusting the alpha level and not the p-values. Another option would be to use Holm’s (or some other) method for p-value adjustment also for the overall p-values, but in our opinion, this would only make the presentation even more difficult to follow without adding much to the validity of the statistical analysis.

3. Reviewer: Please avoid comparison of study results with other studies in the results section. This should only be done in the discussion. This concerns Line 120-121.

Answer: Corrections done. Please, see lines 131-132 and lines 293-294.

4. Reviewer: Line 166: the fact that you compare to the general Slovenian population is not mentioned in methods. Please put this in discussion.

Answer: Corrections done. Please, see lines 293-294 and lines 298-301.

5. Reviewer: Line 169-170: this is already mentioned in methods. Should be in discussion. Not in results and methods.

Answer: Corrections done. Please, see lines 185-188 and lines 377-380.

6. Reviewer: Line 178: please delete the word “trend”

Answer: Corrections done. Please, see line 195.

7. Reviewer: Line 184: is this the corrected P-value?

Answer: Please, see the answer to Results, question No 2.

8. Reviewer: Line 190: “comparably often in all age groups” This is not correct English

Answer: Corrections done. Please, see line 211.

9. Reviewer: Line 199: Kaplan Meier P-value? No survival curves are shown. Please see my suggestion for statistical analysis.

Answer: We believe that showing survival curves would be too space consuming, and we provided the summary of the survival curves in the text. Please, see lines 217-220.

10. Reviewer: Line 200: the large majority: what percent?

Answer: Information added as suggested by the reviewer. Please, see line 221.

11. Reviewer: Line 201: What is the proportion of patients with incomplete response?

Answer: The proportions of patients with incomplete response are provided in Table 2. Please see lines 237-241.

12. Reviewer: Line 204: It is incorrect to say the model was adjusted for.. the model included all these variables. However, in the methods it says that the model was adjusted for age and time for enrollment. How was the model constructed? Was forward or backward selection performed? Or did you just enter all variables of interest? Please consider taking eldery vs other ages as main outcome measure. Table 1 shows univariate analyses of factors associated with age, why did you not perform multivariate analyses on this? Why choose for factors associated with incomplete response only multivariate and not univariate?

Answer: Prior to the analysis it was agreed, based on expert knowledge, which explanatory variables will be considered in the analysis (the ones listed), hence no variable selection was performed. We agree however that this could be stated more explicitly, hence we made changes in the Methods section. Please, see lines 119-120.

13. Reviewer: Line 227: you do not need to put intercept in the table.

Answer: We agree that since the goal was not to build a predictive model, the intercept term can be excluded. Corrections done. Please, see line 250.

14. Reviewer: Line 231: Time from enrollment

Answer: Corrections done. Please, see line 254.

15. Reviewer: Line 239: part of table 3 is missing.

Answer: We believe nothing is missing because OR 7.69 (5.39–10.97) refers to “Presence of LB-associated constitutional symptoms at enrolment (yes vs no)”.

16. Reviewer: Line 245-247: …but none qualified as having “post-Lyme disease syndrome”’, because the reported symptoms did not necessitate reduction of their previous activity levels. This definition should be in the methods. And how was it assessed whether there was a reduction in activity levels.

Answer: We referred to the definition of post-Lyme disease syndrome as it is defined by the Infectious Diseases Society of America (Ref No 13). This definition includes that “subjective symptoms are of such severity that, when present, they result in substantial reduction in previous levels of occupational, educational, social, or personal activities”, which was not the case in any of our patients. The activity levels were not specifically measured, but we believe our conclusion is reasonable nevertheless because none of the patients made any change in his/her occupational, educational, social, or personal activities due to post-LD symptoms. We amended the manuscript as suggested by the reviewer. Corrections done. Please, see line 271, and lines 372-377.

17. Reviewer: Line 252: Possibly in some patients it was not treatment failure but a reinfection. This is noted in the discussion, but could also be mentioned here that there was one case with reinfection, as this is a result.

Answer: Corrections done. Please, see line 276.

18. Reviewer: How many patients had chemiluminescence immunoassay or the C6 ELISA, or immunofluorescence assay? Were ELISA results confirmed with immunoblot according to guidelines?

Answer: Chemiluminescence immunoassay was performed in 817 patients, 284 patients were tested by the immunofluorescence assay, and C6 ELISA was done in 119 patients. Information was added to the manuscript. Please, see lines 173-175. In our hospital we do not perform immunoblot to confirm ELISA when we use the OspC antigen for determining borrelial antibodies which is also recommended by the ECDC.

Discussion

1. Reviewer: Please start the discussion with a summary of your main conclusion. Line 256 is a reference to another paper. Comparison with study results is for second paragraph. Line 260: What are the main differences with respect to age that you found?

Answer: Corrections done. Please, see lines 282-287.

2. Reviewer: Line 278: Possibly in elderly also the time between onset of EM and treatment is longer, which might partly explain the longer duration of resolution of EM.

Answer: Table 1 shows that the time from onset of EM to treatment (line 144) did not differ significantly between the three age groups, therefore we could not provide such an assumption.

3. Reviewer: In patients aged 85 and over you would already expect patients not to be the same as before the event, taking into account comorbidities.

Answer: Actually, this was not the case. There were only three patients aged ≥85 years, therefore we were unable to perform further analysis. Two of these patients attended the 12-month visit and they both qualified as complete responders. The third patient qualified as complete responder at 14 days, but did not attend later follow-up visits. Anyhow, when assessing factors associated with incomplete response (Table 3), comorbidities were taken into account.

4. Reviewer: Line 281:

Answer: /

5. Reviewer: Strenghts of the study are missing in the discussion

Answer: Corrections done. Please, see lines 383-385.

6. Reviewer: Line 318 and 325: rephrase analysis into study

Answer: Corrections done. Please, see lines 372 and 386.

7. Reviewer: Line 333: this part focuses on post-LB symptoms. The aim of the study was to look at age.

Answer: Corrections done. Please, see lines 330-335 and lines 393-397.

Reviewer 3 Report

Boršič and colleagues present their findings on the association between patient age and course of Lyme disease infection among 1220 patients prospectively enrolled and presenting with early Lyme disease with EM.  This is a well-written comprehensive study with important findings.     

Comments:

1.      It is important to emphasize the subjectivity associated with the method used to determine LB-associated constitutional symptoms and post-LB symptoms (i.e. open question about health-related symptoms) and the need for objective measures. 

2.      Please clarify what unidentified means in table 1, line 150. 

3.      The breakdown of the serologic tests that were positive would be useful.  It would also be interesting to know of those who are skin and/or blood culture positive how many patients were positive by serology (and by which test) as compared to those who were not culture positive.        

4.      In the results section, it should be stated upfront that the results for the re-biopsy that tested positive were inconclusive.      

5.      Caution should be used when referring to treatment failure because the results are inconclusive for this group and there is no strong supporting evidence provided for the presence of an active infection in these patients at follow up.    

6.      For the two treatment failures with disease progression, please define how meningitis was diagnosed and how other forms of meningitis were ruled out.  Also, for the patient who developed multiple EMs, was reinfection a possibility? Was culture performed on the new EMs that developed?  What evidence of LB existed in these two patients at enrollment and did they have a previous history of LB?  Were they culture positive at enrollment? Further data and discussion on these two patients would be beneficial for understanding the significance of the results.               

7.      The percentage of young patients enrolled is not the same in the text (line 119) and table 1 (line 125).

Author Response

Author's reply to the Review Report (Reviewer 3)

Comments and Suggestions for Authors

Boršič and colleagues present their findings on the association between patient age and course of Lyme disease infection among 1220 patients prospectively enrolled and presenting with early Lyme disease with EM.  This is a well-written comprehensive study with important findings.     

Comments:

1. Reviewer:  It is important to emphasize the subjectivity associated with the method used to determine LB-associated constitutional symptoms and post-LB symptoms (i.e. open question about health-related symptoms) and the need for objective measures.

Answer: Corrections done as suggested by the reviewer. Please, see line 271 and lines 373-377.

2. Reviewer:  Please clarify what unidentified means in table 1, line 150. 

Answer: Clarification added as suggested. Please, see lines 176-177.

3. Reviewer:  The breakdown of the serologic tests that were positive would be useful.  It would also be interesting to know of those who are skin and/or blood culture positive how many patients were positive by serology (and by which test) as compared to those who were not culture positive.        

Answer: Information added. Please, see lines 173-175 and lines 201-205.

4. Reviewer: In the results section, it should be stated upfront that the results for the re-biopsy that tested positive were inconclusive.      

Answer: Corrections done. Please, see line 276.

5. Reviewer:  Caution should be used when referring to treatment failure because the results are inconclusive for this group and there is no strong supporting evidence provided for the presence of an active infection in these patients at follow up.    

Answer: Information added. Please, see lines 338-339.

6. Reviewer:   For the two treatment failures with disease progression, please define how meningitis was diagnosed and how other forms of meningitis were ruled out. Also, for the patient who developed multiple EMs, was reinfection a possibility? Was culture performed on the new EMs that developed?  What evidence of LB existed in these two patients at enrollment and did they have a previous history of LB?  Were they culture positive at enrollment? Further data and discussion on these two patients would be beneficial for understanding the significance of the results.               

Answer: Additional information added. Please, see lines 346-360.

7. Reviewer:  The percentage of young patients enrolled is not the same in the text (line 119) and table 1 (line 125).

Answer: Corrections done. Please, see line 136.

 Reviewer 4 Report

Some minor comments:

In rows 65-66 it is stated that patients were categorized 18-44, 45-64 and 65 years and over. Was there a special reason for the choice of these cut off ages? For instance, previous data regarding infectious immune responses differing with these age groups, or was the decision only based on opinion? Please, provide a short description on how these categorizes were determined.

In the pdf-file with the manuscript, unfortunately, Tables were not aligned with headings making them difficult to read and interpret, please adjust.

Correct row 125 369/1220 = 30.2% not 30.3%

It seems as information may be missing on some of the 1220 patients regarding certain aspects, e.g. serology, 12 months treatment response and others in Table 1 and 2? Unfortunately, there is no information in the manuscript regarding how many patients completed all study aspects of the original 1220, and a description and short analysis of the “drop-outs”. Please add such a paragraph under Results and a short paragraph of the possible implications on the study results in Discussion. 

Author Response

Author's reply to the Review Report (Reviewer 4)

Comments and Suggestions for Authors

Some minor comments:

1. Reviewer:  In rows 65-66 it is stated that patients were categorized 18-44, 45-64 and 65 years and over. Was there a special reason for the choice of these cut off ages? For instance, previous data regarding infectious immune responses differing with these age groups, or was the decision only based on opinion? Please, provide a short description on how these categorizes were determined.

Answer: Since there is not a universal definition of old because it differs according to the context, we determined age categories based on conclusions drawn from the available categorizations (WHO, Lee CC, Wang JL, Lee CH, et al. Age-Related Trends in Adults with Community-Onset Bacteremia. Antimicrob Agents Chemother. 2017 Nov 22;61(12)). Information was added to the manuscript. Please, see line 69.

2. Reviewer: In the pdf-file with the manuscript, unfortunately, Tables were not aligned with headings making them difficult to read and interpret, please adjust.

Answer: Corrections done as suggested.

3. Reviewer:  Correct row 125 369/1220 = 30.2% not 30.3%

Answer: Corrections done. Please, see line 136.

4. Reviewer:  It seems as information may be missing on some of the 1220 patients regarding certain aspects, e.g. serology, 12 months treatment response and others in Table 1 and 2? Unfortunately, there is no information in the manuscript regarding how many patients completed all study aspects of the original 1220, and a description and short analysis of the “drop-outs”. Please add such a paragraph under Results and a short paragraph of the possible implications on the study results in Discussion.

Answer: Some of the data in Table 1 were provided as n/n (%) due to the reason that some of the characteristics were not available for all patients because not all patients attended all the follow-up visits and/or because not all patients consented for example, to skin biopsy.

In Table 2, we added a column with the numbers of all patients attending each study visit and we also added a line with results of the analysis for the last evaluable visit (where the data for only 3 patients are missing), i.e. "the last observation carry forward approach" and got very similar results as for the 12 months visit. Please, see Table 2, lines 234-241 and lines 268-269.

We also tried the following sensitivity analysis. For each of the patients for which the incomplete response at the final visit was available, transition probabilities from each of the previous visits to the final visit were calculated for each age group. These were then used to randomly generate the outcome for the drop-out patients depending on their age and their incomplete response for their last evaluable visit. The p-values were obtained for the imputed datasets and then this process was repeated 1000 times. The mean p-value across the imputed datasets was 0.09 and 19.1% of the p-values were larger as the one which we calculated on the original data set. Hence, we believe that the drop-out did not substantially bias our results, although it could be assumed that if there was no dropout our obtained effect would be slightly larger and hence the p-value smaller.

However, since this second sensitivity analysis would be difficult to understand for a practicing clinical doctor, we decided to include only the information for the last evaluable visit in the revised manuscript.

Reviewer 5 Report

This manuscript addresses an important clinical and public health question regarding differences in Lyme disease by age. However, much more detail regarding the methodology is required to assess the study design. Please see below.

While authors provide some information about the study sample, much more detail is required. What was the source population from which the participants were recruited? Were they patients seeking care for Lyme disease? Were they primary care patients or specialty care patients? What inclusion/exclusion criteria was applied for each of the clinical trials? What was the participation rate in the study by age? Could their participation in the clinical trials have any impact on their Lyme disease outcomes?

When providing the additional details in patient selection/recruitment, please describe how these methods may have impacted study findings (e.g., participation bias, generalizability).

Provide more details regarding was considered a "constitutional" symptom? Were all symptoms included?

Authors indicate that symptoms were assessed with an "open question about health-related symptoms." What was the question? Was their any prompting? Authors should assess whether it is possible that symptom reporting behavior might differ by age when using an open-ended question. The use of an open-ended question, rather than asking every patient about each symptom, should be noted as a limitation. Recall issues or reporting differences could impact symptom assessment differentially by age and sex.

Author Response

Author's reply to the Review Report (Reviewer 5)

Comments and Suggestions for Authors

This manuscript addresses an important clinical and public health question regarding differences in Lyme disease by age. However, much more detail regarding the methodology is required to assess the study design. Please see below.

1. Reviewer: While authors provide some information about the study sample, much more detail is required. What was the source population from which the participants were recruited? Were they patients seeking care for Lyme disease? Were they primary care patients or specialty care patients? What inclusion/exclusion criteria was applied for each of the clinical trials? What was the participation rate in the study by age? Could their participation in the clinical trials have any impact on their Lyme disease outcomes?

Answer: Explanatory information as suggested by the reviewer was added. Please, see line 58-60 and also answer to question No 2.

2. Reviewer: When providing the additional details in patient selection/recruitment, please describe how these methods may have impacted study findings (e.g., participation bias, generalizability).

Answer: As suggested by the reviewer, we made corrections in the analysis approach. We additionally accounted for the fact that patients from 5 studies were included in the multivariate model by adding a random effect for the trial which in the multivariate analysis adjusts for these differences. The Methods section has been changed accordingly. Please, see lines 122-125, and Table 3, lines 251-261. Note that since all multiple EM patients arise from one trial, including trial as a fixed effect was not possible.

Furthermore: - the main inclusion criteria for the 5 studies were practically the same and differed in two parameters, which we were aware of: one study preferentially included patients with multiple erythema migrans, and in two studies patients with a history of Lyme borreliosis in the past were excluded. We recognized that the uneven sampling might potentially bias our results and adjusted the analysis accordingly, nevertheless we recognized this as one of the study limitations. Please see lines 123-126.

- we did not include treatment regimen as a variable with potential influence on the course and outcome because all patients were treated with antibiotic regimens, which are recommended as efficacious treatment options for treating erythema migrans by the Infectious Diseases Society of America (Ref No 13). Nevertheless, we wrote that different treatment regimens used in the 5 studies were non-inferior to comparators. Corrections done. Please, see lines 75-76.

- the differences in diagnostic methods could not have influenced the results on the course and outcome of EM because no such differences existed. Truly, different serological methods were applied during the study period, however serology was not used as diagnostic criteria. Erythema migrans was defined according to European criteria which do not rely on serological test result. Please, see lines  60-62.

- different serological tests did influence serological results, and we added this information to the manuscript. However, since seropositivity was not used as diagnostic criteria in our study we believe these potential differences did not impact our results on the course and outcome. Furthermore, because each serological method was applied regardless of patients’ age, we believe that differences in serological methods did not impact the direction and strength of correlation between patients’ age and serological results, which was the main question regarding seropositivity in our study. Corrections done. Please, see lines 173-175.

3. Reviewer: Provide more details regarding was considered a "constitutional" symptom? Were all symptoms included?

Answer: All symptoms that had newly developed or worsened since the onset of the EM and which had no other known medical explanation were regarded as LB-associated constitutional symptoms at enrolment or post-LB symptoms at follow-up.

Explanation added. Please, see Line 81-86.

4. Reviewer: Authors indicate that symptoms were assessed with an "open question about health-related symptoms." What was the question? Was their any prompting? Authors should assess whether it is possible that symptom reporting behavior might differ by age when using an open-ended question. The use of an open-ended question, rather than asking every patient about each symptom, should be noted as a limitation. Recall issues or reporting differences could impact symptom assessment differentially by age and sex.

Answer: The information about questioning technique was added to the manuscript. Please, see lines 81-86. As suggested by the reviewer, the possibility that the use of an open-ended question technique had an impact on symptom reporting that may by itself be influenced also by age or sex was added as a limitation of the study. Please, see lines 374-377. 

Round  2

Reviewer 5 Report

The authors adequately addressed the comments provided with the first manuscript review. The only remaining comment is that the discussion section does not need the extensive description of the two patients in whom symptoms progressed during antibiotic treatment (lines 346-360).. The level of detail is not necessary and detracts from the main points of the manuscript.

Author Response

Author's Reply to the Review Report (Reviewer 5), Round 2

Comments and Suggestions for Authors

1. Reviewer:  The authors adequately addressed the comments provided with the first manuscript review. The only remaining comment is that the discussion section does not need the extensive description of the two patients in whom symptoms progressed during antibiotic treatment (lines 346-360). The level of detail is not necessary and detracts from the main points of the manuscript.

Answer: Corrections done. Please, see lines 351-366.